# MISSING BUT NOT LOST: RECONSTRUCTING ARBITRARILY MISSING EEG CHANNELS WITH PRE-TRAINED DIFFUSION

## ABSTRACT

Electroencephalography (EEG) plays a pivotal role in brain–computer interface (BCI) research owing to its non-invasive nature and high temporal resolution. However, the presence of missing channels during acquisition often compromises signal quality and limits practical applications. Existing EEG channel reconstruction methods remain constrained by limited accuracy and generalization. In this work, we present Uni-SVDiffusion, a scalable pre-training–based diffusion framework for cross-channel EEG reconstruction that generates high-fidelity signals for arbitrarily missing channels. Our approach leverages singular value decomposition (SVD) to disentangle EEG signals into spatial and temporal components, and employs a diffusion model trained on spatial representations to achieve precise reconstruction conditioned on observed channels. To enable a unified model across diverse EEG configurations, we further propose a forward/backward channel mapping strategy that preserves spatial structure and facilitates cross-dataset generalization via convolutional pre-training. Evaluations on three datasets demonstrate that Uni-SVDiffusion achieves state-of-the-art performance, even under severe channel-missing scenarios. This work provides a generalizable, plug-and-play solution for EEG reconstruction. Code and pre-trained models will be released upon acceptance.

## 1 INTRODUCTION

Electroencephalography (EEG) has emerged as a widely used modality in brain–computer interface (BCI) research due to its rich representation of neural activity, non-invasive nature, and high temporal resolution (Boonyakitanont et al., 2020; Sharma et al., 2022; Boudewyn et al., 2023). Despite these advantages, practical deployment of EEG is often challenged by data quality issues, most notably channel missing. Reliable EEG acquisition requires stable electrode–scalp contact, yet factors such as subject movement, environmental interference, or individual characteristics (e.g., long or thick hair) frequently degrade connectivity, leading to missing channels. This issue poses significant obstacles for downstream BCI applications, and the reconstruction of missing channel signals remains an open and critical research problem.

Recent studies on EEG reconstruction can be roughly divided into three methodological paradigms. The first relies on traditional mathematical formulations, such as interpolation and matrix factorization (Courellis et al., 2016; Bahador et al., 2021). The second exploits large-scale EEG models, where signal reconstruction is adopted as a pre-training objective to endow the model with the ability to recover EEG signals (Zhang et al., 2023; Yue et al., 2024; Li et al., 2025), which can subsequently be fine-tuned for restoring missing channels in downstream tasks. The third focuses on generative modeling approaches, evolving from early GAN-based frameworks (Lee et al., 2021) to more recent diffusion-based models (Li et al., 2024), which have achieved encouraging results. However, constrained by the limited scale of EEG datasets and the architectural complexity of generative models, these methods remain difficult to pre-train at scale and often fail to generalize across heterogeneous EEG formats.

In this paper, we present a pre-training-based cross-channel generative model for EEG reconstruction that integrates the strengths of prior approaches. Our method first applies singular value de-

composition (SVD) (Golub & Van Loan, 2013) to obtain a compact and unified representation of spatial and temporal features, enabling large-scale pre-training across diverse datasets. A diffusion model is then trained, where temporal features guide the reconstruction of spatial features. Finally, the reconstructed spatial and temporal components are fused to recover complete EEG signals. To enable joint pre-training across datasets with varying numbers and spatial configurations of EEG channels, we propose a channel forward/backward mapping strategy. By predefining a unified set of channels, we map the spatial features of all EEG recordings into a common dimensional space while ensuring that they can be accurately remapped back to their original configurations.

We pre-train our model on nine EEG datasets and perform fine-tuning on three downstream datasets of different tasks. The quality of the generated data is evaluated using four reconstruction metrics. Extensive experimental results demonstrate that our model achieves superior performance compared to existing approaches.

The main contributions of this paper are as follows:

- We introduce a pre-training–based cross-channel EEG generation framework that reconstructs arbitrarily missing channels from partially observed signals.
- We propose a novel forward/backward channel mapping strategy that, in conjunction with SVD, enables a new paradigm of large-scale cross-dataset pre-training for EEG.
- Extensive experiments across multiple datasets demonstrate that our method achieves state-of-the-art performance, and comprehensive analyses further confirm its effectiveness and robustness.

## 2 RELATED WORK

### 2.1 EEG-BASED BCI

Common areas of BCI research include emotion recognition (Houssein et al., 2022), motor imagery (Wang et al., 2024), event-related potentials (Light et al., 2010), and so on. For example, in emotion recognition tasks, participants are typically required to wear an EEG cap while watching various types of videos (such as comedies, tragedies, and horror films) to induce corresponding emotions (Jiang et al., 2024a). In contrast, motor imagery tasks involve participants imagining various motor scenarios upon receiving specific prompts (Liu et al., 2024a). In recent years, there have been more novel studies on EEG, such as recording the EEG signals of participants while watching pictures or videos, and attempting to decode them to reconstruct the visual stimulus material watched by the participants (Ahmed et al., 2021; Liu et al., 2024b). All of the above studies require high-quality EEG data as a prerequisite. In a laboratory environment, we can minimize external interference as much as possible by setting up a good experimental environment, while keeping the subjects as comfortable as possible to obtain high-quality EEG signals. However, in practical applications, there are many uncontrollable factors in EEG acquisition scenarios, making it difficult to ensure signal quality. This problem also poses a great obstacle to the applications of non-invasive BCI.

### 2.2 EEG GENERATION MODEL

Early methods for generating EEG channels focused on statistical learning methods (Courellis et al., 2016). Bahador *et al.* proposed a method to repair the EEG by calculating the correlation between channels and using the weighted sum of visible channels (Bahador et al., 2021). Akmal *et al.* employed tensor-based interpolation methods to reconstruct missing EEG data (Akmal et al., 2021). Such interpolation-based methods struggle to model the nonlinear and time-varying nature of EEG, resulting in inherent limitations in reconstruction quality. Generative model-based EEG reconstruction methods have achieved notable progress. Lee *et al.* proposed a method based on Generative Adversarial Networks (GAN) to generate realistic EEG signal sequences for sleep EEG data (Lee et al., 2021). Li *et al.* proposed a diffusion model-based approach for biomedical signal generation (Li et al., 2024). However, these methods do not account for channel mismatches across datasets, making the trained models difficult to generalize to new EEG data. Although reconstruction methods based on large pretrained EEG models enable cross-dataset training and fine-tuning, their performance is often inferior to that of dedicated reconstruction models, as they are not specifically designed for channel restoration.

## 3 METHOD

### 3.1 PRELIMINARY

We introduce some definitions and properties of SVD in this subsection, which will be used later.

**Theorem 3.1** *Assume $A = (a_{ij}) \in \mathbb{C}^{m \times n}$, and $\sigma_1 \geqslant \sigma_2 \geqslant \cdots \geqslant \sigma_r > 0$, then there exist unitary matrices $U$ of order $m$ and $V$ of order $n$ such that:*

$$A = UDV^*, \tag{1}$$

*where $D = diag(\sigma_1, \sigma_2, \ldots, \sigma_r, 0, \ldots, 0)_{m \times n}$.*

The formula 1 is referred to as the singular value decomposition (SVD) of matrix $A$, and $\sigma_1, \sigma_2, \ldots, \sigma_r, 0, \ldots, 0$ (a total of n values) are called the singular values of $A$.

The SVD of a matrix has the following properties:

**Proposition 3.2** *Assume the rank of $A$ is $r$, The first $r$ columns of matrix $U$ form an orthonormal basis for the column space of A, and the first $r$ rows of matrix $V^*$ form an orthonormal basis for the row space of A.*

**Proposition 3.3** *When $m < n$ and the matrix $A$ is full-row rank, the number of singular values equals the number of rows of A, i.e., $r = m$.*

**Proposition 3.4** *The product of the matrix $V_k$, formed by the first $k$ rows of matrix $V^*$, and its transpose is the identity matrix, i.e., $V_k V_k^* = I_k$, where $I_k$ represents the $k \times k$ identity matrix.*

### 3.2 PROBLEM FORMULATION

Consider the EEG signal $X \in \mathbb{R}^{C \times T}$, where $C$ is the channel (electrode) number and $T$ is the samples in time dimension. We define the phenomenon of channel missing as the scenario where the data with length $T$ from certain channels are completely unavailable. The visible and missing EEG data are denoted as $X_V \in \mathbb{R}^{C_V \times T}$ and $X_M \in \mathbb{R}^{C_M \times T}$, respectively, where $C_V$ and $C_M$ are the number of visible channels and missing channels, and $C_V + C_M = C$. The missing ratio $\eta$ is defined as:

$$\eta = C_M/C. \tag{2}$$

During training, we assume all EEG channels are available, i.e., $X_V = X$. In the testing phase, EEG data experiences channel missing at a certain $\eta$ value, and our goal is to reconstruct $X_M$ with the guidance of $X_V$.

### 3.3 MOTIVATION

According to proposition 3.2, the row vectors of $V^*$ reflect the characteristics of the row space of $A$. In the case of EEG signals, the row space corresponds to the temporal features, which means $V^*$ is closed associated with the temporal features of the EEG signals. Similarly, $U$ is closely related to the spatial features of the EEG signals.

A key point in reconstructing the missing channels of the EEG signal is how to utilize the visible portion of the EEG data. According to the definition of SVD, we have noticed that regardless of how many EEG channels are retained, as long as the number of time samples is consistent, the matrix $V^*$ obtained from the SVD will always have a fixed size of $T \times T$. Since the $V$ matrix obtained from the SVD of both $X$ and $X_V$ has the same format and is closely related to the temporal features, can we share this characteristic and focus solely on reconstructing the spatial features of the EEG? Based on this idea, we have designed the **Uni-SVDiffusion** model.

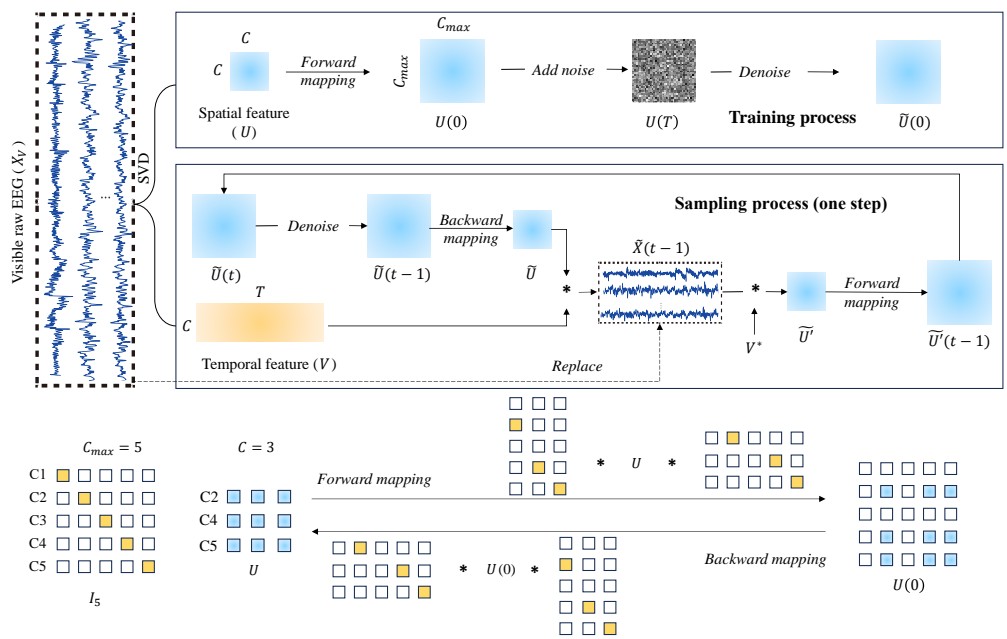

Figure 1: The overall architecture of Uni-SVDiffusion. During the training phase, the visible raw EEG comes from all channels. During the sampling phase, only partial channel data is available, and the temporal features from these channels are utilized to guide the sampling process. The lower part of the figure illustrates how forward/backward mapping is used to unify EEG data with different channel numbers into a consistent format and to reconstruct it.

## 3.4 ARCHITECTURE

### 3.4.1 TRAINING BASED ON SVD

The overall architecture of our method is shown in Figure 1. During training, we have complete EEG data $X \in \mathbb{R}^{C \times T}$. According to the definition of SVD, it can be decomposed as:

$$X = U_X D V_X, \tag{3}$$

where $U_X \in \mathbb{R}^{C \times C}$, $D \in \mathbb{R}^{C \times T}$, and $V_X \in \mathbb{R}^{T \times T}$. Since EEG signals exhibit nearly random variations over time and there is no direct connectivity between electrodes, when the number of samples in an EEG signal exceeds the number of channels, it is almost impossible for there to be a linear relationship between the signals of different channels. This ensures that each segment of EEG data is full-row rank. So, according to the proposition 3.3, the number of singular values of $X$ equals $C$. We can simplify the above equation as follows:

$$X = U_X D V_X = UV, \tag{4}$$

where $U \in \mathbb{R}^{C \times C}$ is the first $C$ columns of $U_X D$ and $V \in \mathbb{R}^{C \times T}$ is the first $C$ rows of matrix $V_X^*$.

According to the proposition 3.4, we multiply both sides of Equation 4 by $V^*$:

$$XV^* = UVV^* = UI_C = U. \tag{5}$$

In this way, we can represent the spatial features of EEG using the visible raw signals and temporal features. During the pre-training phase, we train a diffusion model on these spatial features.

A key challenge lies in the fact that the features $U$ extracted from different datasets differ not only in the number of channels but also in their feature dimensions. Consequently, conventional Transformer-based EEG pre-training methods are not applicable due to the inability to downsample and align dimensions (Yang et al., 2023; Jiang et al., 2024b). Similarly, it is infeasible to construct a convolutional network with a unified number of channels, making joint pretraining unattainable. To address this issue, we propose a forward & backward mapping strategy for the EEG channels.

### 3.4.2 FORWARD & BACKWARD MAPPING STRATEGY

To enable joint training of EEG spatial features with inconsistent channels and dimensions, we propose a forward & backward mapping strategy. We define an identity matrix $I_{C_{max}}$, where $C_{max}$ is the maximum number of channels the diffusion model can accommodate, and each row of this identity matrix corresponds to a channel. Take the lower part of the Figure 1 as an example. We predefine a $5 \times 5$ identity matrix $I_5(C_{max} = 5)$, where each row corresponds to one of the five channels, C1 through C5. Suppose a particular EEG dataset comprises channels C2, C4, and C5, whose spatial feature $U$ can be denoted as a $3 \times 3$ matrix ($C = 3$). We extract the row vectors from $I_5$ corresponding to these three channels and assemble them into a new matrix $P$. It is worth noting that:

$$PP^* = I_3. \tag{6}$$

Then, the three-channel $U$ data can be expanded to a five-channel representation and subsequently reconstructed in its original form using the following approach:

$$U \xrightleftharpoons[PU(0)P^*]{P^*UP} U(0). \tag{7}$$

Similarly, EEG data comprising any subset of channels within C1 to C5 can be expanded to a five-channel representation and accurately remapped using the same approach. This enables datasets with differing channel configurations to be jointly trained within a unified framework.

Theoretically, this approach allows us to define $C_{max}$ as a large number to encompass all possible EEG channels. In practical experiments, however, considering that most publicly available EEG datasets contain only a few dozen channels, we set $C_{max}$ to 95, which covers the vast majority of commonly used EEG channels. We can now employ a one-dimensional convolutional U-Net with a fixed initial channel size of 95 to train data from different datasets.

**P.S.** Two important details should be noted during the training process.

- When different datasets share the same channel names and numbers but differ in channel order, directly applying the above mapping method may lead to inconsistent mappings. Therefore, it is necessary to first reorder the channels in each dataset according to the predefined order specified in the identity matrix $I_{C_{max}}$ (See Appendix A.5).

- We recommend using the L1 loss function for model training, the reason can be found in Section 4.5.1.

### 3.4.3 SAMPLING WITH THE GUIDANCE

When channel missing occurs, $X_V \neq X$. We perform SVD on $X_V$, and utilize the top $C$ rows of $V_{X_V}$ (denoted as $V$) to guide the diffusion model in sampling, thereby obtaining the complete spatial features $U$. Subsequently, we integrate the two to reconstruct the full EEG data.

We input an initialized Gaussian noise $\tilde{U}$ of size $C_{max} \times C_{max}$ and denoise it step by step. Take the process within the 'Sampling process' in Figure 1 as an example. At each step of denoising, for $\tilde{U}(t)$ at time t, we first denoise it to obtain $\tilde{U}(t-1)$, then we perform backward mapping to extract the spatial features corresponding to the complete EEG channels, and multiply them with the temporal features of the visible EEG to obtain the pseudo-EEG data $\tilde{X}(t-1) \in \mathbb{R}^{C \times T}$.

In order to ensure that the sampling results are as consistent as possible with the real signal, we replace the data of the corresponding channel in $\tilde{X}(t-1)$ with the value of $X_V$, and then recalculate the spatial features $\tilde{U}'$ as:

$$\tilde{U}' = \tilde{X}(t-1)V^*, \tag{8}$$

where $V^*$ denotes the transpose of the matrix $V$. It is important to note that when performing Equation 8, the channels replaced in $\tilde{X}(t-1)$ do not affect the computation results of other channels. In other words, the row vectors corresponding to the missing channel positions in $\tilde{U}'$ are identical to the values at the same positions in $\tilde{U}$, which is determined by the properties of the inverse matrix operation. Similarly, we apply a forward mapping to $\tilde{U}'$ again, transforming it into a matrix $\tilde{U}'(t-1) \in \mathbb{R}^{C_{max} \times C_{max}}$.

At this point, the row vectors in $\tilde{U}'(t-1)$ corresponding to the visible channels are the true results we aim to achieve through the denoising of $\tilde{U}(t)$, and then $\tilde{U}'(t-1)$ is applied in the next step of sampling and denoising. By keeping a portion of the data as the true values during each sampling step, we hope to guide the row vectors of the other channels to converge toward the desired target.

After multiple steps of denoising, we use the matrix $\tilde{U}$ obtained at the final step as the true spatial feature $U$ of the complete EEG data, and the complete reconstructed EEG $\tilde{X}$ is calculated as:

$$\tilde{X} = \tilde{U}'V. \tag{9}$$

## 4 EXPERIMENTS

### 4.1 DATASETS

We pre-train our model on nine datasets derived from different tasks and subsequently fine-tune it on three downstream tasks. The basic information of all datasets is summarized in Table 1. More detailed description of datasets and their channels can be found in Appendix A.2. Some datasets already undergo preprocessing, and we preserve their original formats. For the raw data that are not preprocessed, we apply a band-pass filter with a frequency range of 0.1–75 Hz to remove low-frequency drifts and high-frequency noise. All datasets are publicly available and can be accessed online.

Table 1: The basic information of all datasets used in the experiment. 'Duration' refers to the total recording time of all samples in each dataset (hours).

| Stage | Name | Channel | Sampling Rate (Hz) | Subject | Sample/Duration (h) |
|---|---|---|---|---|---|
| Pre-train | Convert Shifts of Attention (Treder et al., 2011a) | 60 | 200 | 8 | 20842/5.8 |
| | Center Speller (Treder et al., 2011b) | 63 | 250 | 13 | 33911/9.4 |
| | Target Versus Non Target (bi2015a) (Korczowski et al., 2019) | 32 | 512 | 43 | 41928/11.6 |
| | Individual Imagery (Scherer et al., 2015) | 30 | 256 | 9 | 47351/13.2 |
| | Music BCI (Treder et al., 2014) | 63 | 200 | 11 | 121325/33.7 |
| | RSVP Speller (Acqualagna & Blankertz, 2013) | 63 | 200 | 12 | 26030/7.2 |
| | SEED-IV (Zheng et al., 2018) | 62 | 200 | 15 | 105165/29.2 |
| | SEED-VII (Jiang et al., 2024a) | 62 | 200 | 20 | 280079/77.8 |
| | SHU (Ma et al., 2022) | 32 | 250 | 25 | 239760/66.6 |
| Fine-tune | SEED (Duan et al., 2013) | 62 | 200 | 15 | 90450/- |
| | Monitoring ERP (Chavarriaga & Millán, 2010) | 64 | 512 | 6 | 8285/- |
| | BCI Competition IV-2a (Motor imagery) (Brunner et al., 2008) | 22 | 250 | 9 | 10368/- |

### 4.2 EXPERIMENTAL DETAILS

During pre-training and fine-tuning, we select a one-second duration as a sample, meaning that $T$ equals the sampling rate of each dataset. We perform standard normalization on each sample of the EEG data separately for each channel, where even if some channels are missing, the normalization of the visible channels will remain unaffected. During testing, a portion of the channels in each sample is randomly missing according to the specified channel dropout rate. For the channel missing ratio, we select three values: 0.1, 0.3, and 0.5. The model comprises approximately 20M parameters (19693407 in total).

For baseline selection, we choose the most representative and up-to-date models from the three categories of reconstruction approaches discussed earlier. Specifically, the baselines include: spherical spline interpolation (SSI), which best conforms to the head geometry (Perrin et al., 1989); two reconstruction methods based on diffusion models (RePaint (Lugmayr et al., 2022) and BioDiffusion (Li et al., 2024)); and Gram, the largest publicly available EEG foundation model to date, which employs a Transformer architecture for raw EEG signal reconstruction (Li et al., 2025). Appendix A.3 shows more detailed baseline information.

To evaluate the reconstruction performance of these models, we employ four metrics: Pearson Correlation Coefficient (PCC) (Pearson, 1896), Relative Root Mean Squared Error (RRMSE), Peak Signal-to-Noise Ratio (PSNR), and Structural Similarity Index Measure (SSIM) (Wang et al., 2004). Since the last two metrics are primarily used in the image domain, we make appropriate modifications to adapt them for evaluating EEG signals. Given the original EEG data $X$ and the reconstructed

Table 2: The results (mean/standard deviation) of different methods on the three datasets. [%] indicates that the metric is expressed as a percentage for clarity in presentation. The best results are highlighted in bold with a pink background, while the suboptimal results are shown with a yellow background.

| Dataset | Method | 0.1 ($\eta$) | | | | 0.3 ($\eta$) | | | | 0.5 ($\eta$) | | | |
|---|---|---|---|---|---|---|---|---|---|---|---|---|---|
| | | PCC [%](↑) | RRMSE [%](↓) | PSNR (↑) | SSIM [%](↑) | PCC [%](↑) | RRMSE [%](↓) | PSNR (↑) | SSIM [%](↑) | PCC [%](↑) | RRMSE [%](↓) | PSNR (↑) | SSIM [%](↑) |
| SEED | SSI | 56.98/3.98 | 88.92/6.43 | 1.44/2.77 | 25.38/7.17 | 56.02/7.49 | 92.16/7.41 | 0.85/0.58 | 24.74/5.52 | 35.38/6.05 | 112.85/3.78 | -0.99/1.04 | 10.99/5.92 |
| | RePaint | 57.65/6.93 | 84.25/2.90 | 1.49/0.30 | 11.61/2.82 | 49.35/5.92 | 87.23/2.50 | 1.19/0.25 | 9.03/2.07 | 40.39/5.04 | 90.97/2.16 | 0.82/0.21 | 6.51/1.55 |
| | BioDiffusion | -3.31/1.44 | 109.35/0.83 | -0.78/0.07 | 0.25/0.03 | -2.98/1.26 | 109.23/0.76 | -0.77/0.06 | 0.25/0.03 | -2.60/1.10 | 109.15/0.70 | -0.76/0.06 | 0.25/0.02 |
| | **Uni-SVDiffusion (Ours)** | **82.34/5.26** | **55.10/6.43** | **5.23/1.00** | **45.16/6.12** | **80.62/5.47** | **57.78/6.15** | **4.81/0.91** | **41.67/5.88** | **76.75/5.65** | **63.25/5.40** | **4.01/0.73** | **35.08/5.14** |
| | Uni-SVDiffusion (w/o fine-tune) | 80.15/5.75 | 58.74/7.04 | 4.68/1.01 | 41.35/6.31 | 78.28/5.94 | 61.28/6.63 | 4.30/0.91 | 38.14/5.93 | 74.11/5.91 | 66.37/5.64 | 3.59/0.71 | 32.12/4.97 |
| Monitor ERP | SSI | 86.64/5.96 | 47.49/8.36 | 6.58/2.55 | 40.09/4.59 | 81.47/11.62 | 56.75/7.02 | 4.83/1.35 | 35.68/8.01 | 63.19/9.76 | 81.09/8.02 | 2.59/1.05 | 21.23/6.97 |
| | RePaint | 78.48/5.37 | 66.24/5.40 | 3.61/0.72 | 22.80/6.01 | 77.09/4.68 | 67.69/4.62 | 3.41/0.62 | 22.30/5.03 | 72.48/4.72 | 72.40/4.62 | 2.82/0.57 | 18.46/5.09 |
| | BioDiffusion | 24.54/13.21 | 101.51/7.09 | -0.11/0.61 | 3.77/2.63 | 25.17/13.09 | 101.18/7.09 | -0.08/0.61 | 3.89/2.63 | 27.14/13.32 | 100.12/7.35 | 0.01/0.64 | 4.14/2.75 |
| | Gram | 34.27/14.09 | 99.65/0.15 | 0.03/0.01 | 0.02/0.01 | 34.86/14.25 | 99.65/0.15 | 0.03/0.01 | 0.02/0.01 | 35.34/14.49 | 99.65/0.15 | 0.03/0.01 | 0.02/0.01 |
| | **Uni-SVDiffusion (Ours)** | **89.46/4.46** | **43.65/9.19** | **7.41/1.96** | **46.23/4.70** | **88.59/4.68** | **46.38/8.81** | **6.84/1.73** | **43.20/4.50** | **86.69/5.04** | **52.18/7.77** | **5.75/1.32** | **37.66/4.12** |
| | Uni-SVDiffusion (w/o fine-tune) | 84.60/4.64 | 53.23/10.68 | 5.67/1.85 | 37.82/4.59 | 83.65/7.14 | 56.45/9.97 | 5.11/1.61 | 34.61/4.24 | 81.87/7.82 | 61.76/8.37 | 4.27/1.22 | 29.61/3.49 |
| Motor imagery | SSI | 92.14/1.00 | 28.89/5.15 | 9.67/2.14 | 70.02/5.47 | 90.44/1.63 | 37.12/6.82 | 7.11/2.20 | 60.67/5.33 | 80.66/8.90 | 62.54/4.64 | 4.18/1.78 | 50.02/5.97 |
| | RePaint | 91.08/01.59 | 64.35/02.22 | 3.83/0.30 | 36.80/3.22 | 83.14/1.70 | 72.54/1.37 | 2.79/0.16 | 24.57/1.83 | 67.42/1.89 | 80.64/0.98 | 1.87/0.11 | 14.75/1.09 |
| | BioDiffusion | 6.84/2.49 | 103.19/0.90 | -0.27/0.08 | 0.25/0.05 | 6.78/2.61 | 103.20/0.93 | -0.27/0.08 | 0.26/0.06 | 6.36/2.59 | 103.36/0.95 | -0.29/0.08 | 0.26/0.06 |
| | Gram | 84.36/8.37 | 59.43/9.20 | 4.65/1.29 | 37.77/7.01 | 84.07/8.28 | 60.01/8.97 | 4.55/1.24 | 37.04/6.79 | 83.68/8.16 | 60.75/8.57 | 4.44/1.17 | 35.91/6.35 |
| | **Uni-SVDiffusion (Ours)** | **95.72/1.14** | **28.99/3.63** | **10.83/1.12** | **71.90/2.99** | **95.32/1.20** | **30.08/3.81** | **10.51/1.14** | **70.57/3.20** | **94.41/1.34** | **32.84/3.85** | **9.73/1.06** | **67.26/3.47** |
| | Uni-SVDiffusion (w/o fine-tune) | 90.56/1.19 | 61.41/0.92 | 4.24/0.13 | 41.03/1.69 | 87.09/1.17 | 65.92/0.82 | 3.62/0.11 | 35.53/1.53 | 80.27/1.15 | 72.89/0.71 | 2.75/0.09 | 27.35/1.39 |

EEG data $\tilde{X}$, the specific definitions of both are as follows:

$$PSNR = 10 \cdot log_{10}(\frac{||X||_2^2}{||X - \tilde{X}||_2^2}), \qquad SSIM = \frac{(2\mu_X \mu_{\tilde{X}} + C_1)(2\sigma_{X\tilde{X}} + C_2)}{(\mu_X^2 + \mu_{\tilde{X}}^2 + C_1)(\sigma_X^2 + \sigma_{\tilde{X}}^2 + C_2)}, \qquad (10)$$

where $\mu$ and $\sigma^2$ represent the mean and variance of the corresponding data, respectively, $\sigma_{X\tilde{X}}$ is the covariance between the two data, and $C_1$ and $C_2$ are constants used to stabilize the computation.

## 4.3 RESULTS

The main results on the three datasets are shown in Table 2. The Gram model is excluded from the reconstruction evaluation on the SEED dataset, as it was pre-trained on the entire SEED dataset, rendering a fair comparison invalid. As shown in the table, our model consistently achieves superior performance across nearly all scenarios, regardless of whether the dataset involves multiple channels (64/62) or fewer channels (15).

It is noteworthy that our model achieves strong performance even without fine-tuning, demonstrating robust zero-shot capability when directly evaluated with pre-trained weights. To further assess the validity of the reconstructed data, we conduct classification experiments on the SEED dataset: both the original and reconstructed data are tested using the same pre-trained classifiers (SPaRCNet (Jing et al., 2023), ContraWR (Yang et al., 2021), and CNNTrans (Peh et al., 2022)). As shown in Figure 2(A), the reconstructed data from our method exhibits the smallest drop in classification accuracy, indicating superior preservation of fidelity and discriminative information.

## 4.4 ANALYSIS

In this section, we analyze the model from two perspectives: the impact of forward & backward mappings on the model and the model's ability to generate arbitrary-length data.

### 4.4.1 FORWARD & BACKWARD MAPPING

To enable different EEG data to share the same model, we adopt a channel mapping strategy. However, this operation leads to a sparse spatial feature matrix $U$. To investigate whether it affects model training, we compare the following three models: 1) the same diffusion model trained directly without pre-training but using channel mapping to process $U$ (**w/o pre-train**); 2) the model trained directly without either pre-training or channel mapping (**w/o mapping**); and 3) our Uni-SVDiffusion model. Figure 2(B) shows partial reconstruction metrics on three datasets.

From the figure, it can be observed that, without large-scale pre-training, performing forward/backward mapping on the data leads to a noticeable degradation in reconstruction quality. Since the model architecture and training configurations are held constant, this performance drop can be attributed to the introduction of sparse matrices. In contrast, after applying large-scale pre-training, the model exhibits substantial performance improvements. Even with the incorporation of sparse matrices, it achieves the best results, which highlights both the effectiveness and necessity of large-scale pre-training in our model. More results can be found in Appendix A.6.1.

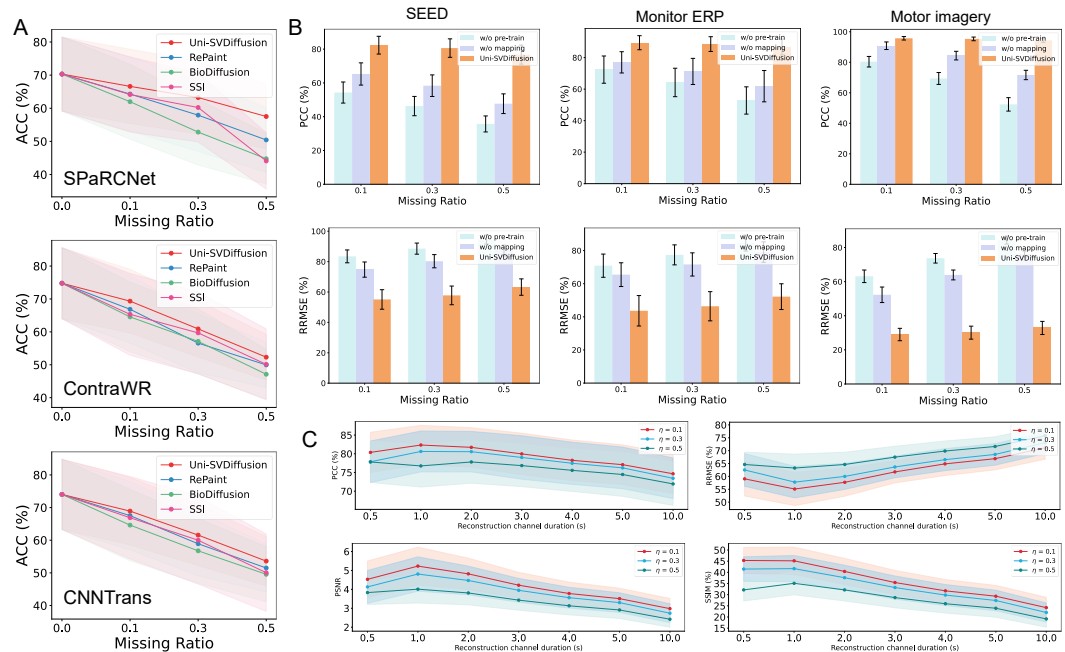

Figure 2: (A) Classification accuracy of reconstructed EEG signals using different methods on three classifier models. (B) The impact of the channel mapping strategy on model performance. (C) Trends in reconstruction metrics across inputs of arbitrary lengths.

Table 3: The results (mean/standard deviation) of the loss function ablation study of Uni-SVDiffusion on three datasets, with the best results highlighted in bold.

| Dataset | Loss | 0.1 ($\eta$) | | | | 0.3 ($\eta$) | | | | 0.5 ($\eta$) | | | |
|---|---|---|---|---|---|---|---|---|---|---|---|---|---|
| | | PCC [%]($\uparrow$) | RRMSE [%]($\downarrow$) | PSNR ($\uparrow$) | SSIM [%]($\uparrow$) | PCC [%]($\uparrow$) | RRMSE [%]($\downarrow$) | PSNR ($\uparrow$) | SSIM [%]($\uparrow$) | PCC [%]($\uparrow$) | RRMSE [%]($\downarrow$) | PSNR ($\uparrow$) | SSIM [%]($\uparrow$) |
| SEED | L1 | **82.34/5.26** | **55.10/6.43** | **5.23/1.00** | **45.16/6.12** | **80.62/5.47** | **57.78/6.15** | **4.81/0.91** | **41.67/5.88** | **76.75/5.65** | **63.25/5.40** | **4.01/0.73** | **35.08/5.14** |
| | L2 | 68.42/6.55 | 71.83/5.51 | 2.90/0.66 | 25.91/4.77 | 62.47/6.43 | 77.22/4.83 | 2.26/0.53 | 20.40/3.82 | 52.50/5.82 | 84.60/3.87 | 1.46/0.39 | 13.85/2.54 |
| Monitor ERP | L1 | **89.46/4.46** | **43.65/9.19** | **7.41/1.96** | **46.23/4.70** | **88.59/4.68** | **46.38/8.81** | **6.84/1.73** | **43.20/4.50** | **86.69/5.04** | **52.18/7.77** | **5.75/1.32** | **37.66/4.12** |
| | L2 | 83.70/5.97 | 57.70/7.20 | 4.85/1.11 | 31.31/4.99 | 79.46/6.77 | 64.04/6.88 | 3.92/0.98 | 25.53/5.19 | 72.11/8.67 | 72.15/6.44 | 2.87/0.82 | 18.78/4.54 |
| Monitor imagery | L1 | **95.72/1.14** | **28.99/3.63** | **10.83/1.12** | **71.90/2.99** | **95.32/1.20** | **30.08/3.81** | **10.51/1.14** | **70.57/3.20** | **94.41/1.34** | **32.84/3.85** | **9.73/1.06** | **67.26/3.47** |
| | L2 | 95.36/1.20 | 31.83/3.66 | 10.00/0.99 | 68.57/3.25 | 94.53/1.30 | 35.26/3.72 | 9.10/0.92 | 64.99/3.68 | 92.75/1.64 | 43.67/3.60 | 7.23/0.72 | 56.53/4.23 |

### 4.4.2 RECONSTRUCT DATA OF ARBITRARY LENGTH

The pre-training, fine-tuning, and testing procedures described above are all conducted using 1-second data segments. Considering that the duration of data to be reconstructed in practical applications is not fixed, and that one of the major advantages of our model lies in its ability to process input data without constraints on the temporal dimension, we evaluate the performance of Uni-SVDiffusion in reconstructing EEG signals of arbitrary length in a single pass and analyze the variations in reconstruction metrics. We conduct experiments on the SEED dataset, which contains longer individual trials. As shown in Figure 2(C), as the length of data reconstructed in a single pass increases, the reconstruction performance of the model gradually declines; however, the degradation follows a smooth trend and remains within an acceptable range. When the reconstructed segments are relatively short, the reconstruction quality shows little to no noticeable variation.

### 4.5 ABLATION STUDIES

#### 4.5.1 ABLATION OF LOSS FUNCTION

In the above experiments, we use the L1 loss function. We find that replacing it with the L2 loss function leads to a significant performance drop, as shown in Table 3. Moreover, this discrepancy is not limited to our method; similar issues arise with other diffusion-based baselines as well (See Table 5 in Appendix A.6.2).

We note that both Chen *et al.* and Saharia *et al.* recommend the use of the L1 loss function in their studies (Chen et al., 2020; Saharia et al., 2022b). The former focuses on generating high-

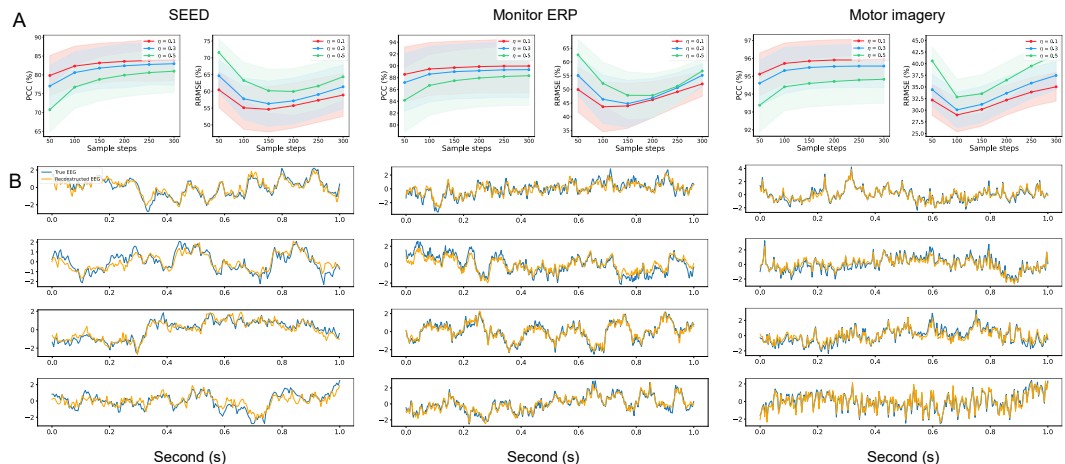

Figure 3: Ablation study results and visualizations. (A) Partial results of the ablation study on sampling steps; (B) Visualization of reconstructed EEG channels on three datasets.

quality audio, while the latter aims to enhance image resolution. Both tasks require output to closely match real reference samples. In a follow-up study, Saharia *et al.* further find that models trained with L2 loss exhibit greater sample diversity compared to those trained with L1 loss (Saharia et al., 2022a). Considering these studies, we believe that for reconstruction tasks, the primary objective is to faithfully recover the original data rather than to promote diversity. Therefore, the L1 loss is more suitable in this context.

### 4.5.2 ABLATION OF SAMPLING STEPS

We investigate the impact of the number of sampling steps on reconstruction performance. As shown in Figure 3(A), as the number of sampling steps increases, the PCC metric gradually rises and eventually stabilizes, whereas the RRMSE metric exhibits a clear initial decrease followed by an increase. This indicates that although reconstructions obtained with larger sampling steps closely follow the overall trends of the ground truth signals, the details in more complex regions are increasingly distorted, leading to higher RRMSE values. More complete results and the visualization of this phenomenon are provided in Figure 7 of Appendix A.6.4.

### 4.6 VISUALIZATION

Figure 3(B) shows the visualization of reconstructed signals on three datasets ($\eta = 0.3$), the blue curves represent the original EEG signals, whereas the yellow curves illustrate the signals reconstructed by our model. Each small panel depicts one second of data. More visualizations can be found in Appendix A.6.5.

## 5 CONCLUSION

In this paper, we propose Uni-SVDiffusion, a pretrained cross-channel EEG generation model. By incorporating SVD and forward/backward mapping, we enable cross-dataset pretraining of EEG diffusion models. Guided by partially observed channels, our model can generate EEG signals at unobserved locations, enabling missing data reconstruction and enhancing the spatial resolution of EEG signals.

**Large Language Model:** In this work, we employ a large language model (ChatGPT) to refine certain sentences, making them more aligned with academic and professional expressions.

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

## A  APPENDIX

### A.1  THE FEASIBILITY ANALYSIS OF USING SVD FOR EEG SIGNAL PROCESSING.

Given its significance in matrix analysis, applying SVD to EEG signal processing is a natural approach (Sadasivan & Dutt, 1996; Hassanpour et al., 2004). This section evaluates the appropriateness of using SVD in our method, considering both data format and computational complexity.

The core idea of our model is to separate the temporal and spatial features of EEG signals, so that even when there are missing channels and incomplete spatial features in EEG, the shared temporal features can still be used to guide the reconstruction process. In theory, all EEG signals $X \in \mathbb{R}^{C \times T}$ can be decomposed into spatially and temporally correlated matrices $U$ and $V$ through SVD. When $C \leqslant T$, we have demonstrated in the paper that by multiplying the left singular matrix $U$ with the singular value matrix $D$, a $C \times C$ format spatial feature matrix can be simplified. However, when $C > T$, the maximum number of singular values of $X$ is $T$, which is less than $C$. The product of $U$ and $D$ is a matrix in the $C \times T$ format. Although the diffusion model can still be used for denoising, its computational complexity is comparable to directly processing the original signal, and it cannot fully utilize the advantages of our model. Moreover, during sampling, if the time length changes, the noise size will also change, and the denoising effect cannot be guaranteed. So our model is

more suitable for EEG signals in $C \leqslant T$ format. The current mainstream EEG acquisition devices are mostly 32 to 64 channels, with a sampling rate of several hundred hertz, which fully meets our requirements. For some high-density EEG acquisition devices, we can increase the length of the training samples appropriately to ensure $C \leqslant T$,

The computational complexity of performing SVD on an $m \times n$ matrix is typically $O(min(mn^2, m^2n))$. For EEG signals, as mentioned earlier, the number of channels is typically in the range of tens, which is much smaller than the sampling rates of several hundred or even thousands Hz. Therefore, in this context, the complexity is determined by the number of channels and can be expressed as $O(C^2T)$. Although this still represents an approximate cubic computation, the upper limit of $C$ and the ability to adjust the time segments ensure a reasonable computational speed. As for spatial complexity, modern computing systems are fully capable of handling the $O(C^2)$ spatial complexity (since we only use $U$ as training data). Based on the above, any EEG signal can meet our requirements.

## A.2 DATASETS INFORMATION

The datasets used for model pre-training and fine-tuning are listed below:

- **Convert Shifts of Attention**: This dataset is designed to investigate whether different directions of attentional shifts can be reliably distinguished based on EEG signals. To this end, eight healthy participants were instructed to maintain strict central fixation while covertly shifting their visual attention to one of six cued directions.

  *60 channels: Iz, I1, AF3, AF4, I2, F7, F5, F3, F1, Fz, F2, F4, F6, F8, PO9, FC5, FC3, FC1, FCz, FC2, FC4, FC6, PO10, T7, C5, C3, C1, Cz, C2, C4, C6, T8, TP7, CP5, CP3, CP1, CPz, CP2, CP4, CP6, TP8, P9, P7, P5, P3, P1, Pz, P2, P4, P6, P8, P10, PO7, PO3, POz, PO4, PO8, O1, Oz, O2.*

- **Center Speller**: The aim of this dataset is to develop a visual speller that operates without relying on eye movements. In an online experiment, thirteen healthy participants tested three different variants of a two-stage visual speller based on covert spatial attention and non-spatial feature-based attention (i.e., attention to color and shape). In the first stage, a group of letters was selected; in the second stage, the desired letter was chosen from within the selected group. The central speller employed distinct geometric shapes, each uniquely associated with a specific color. These shapes were presented sequentially at the center of the screen.

  *63 channels: Iz, Fp2, I1, AF3, AF4, I2, F9, F7, F5, F3, F1, Fz, F2, F4, F6, F8, F10, PO9, FC5, FC3, FC1, FCz, FC2, FC4, FC6, PO10, T7, C5, C3, C1, Cz, ,C2, C4, C6, T8, TP7, CP5, CP3, CP1, CPz, CP2, CP4, CP6, TP8, P9, P7, P5, P3, P1, Pz, P2, P4, P6, P8, P10, PO7, PO3, POz, PO4, PO8, O1, Oz, O2.*

- **Target Versus Non Target (bi2015a)**: This dataset contains EEG recordings of 50 subjects (only 43 subjects' data are used in our experiment) playing to a visual P300 BCI videogame named Brain Invaders. The interface uses the oddball paradigm on a grid of 36 symbols (1 Target, 35 Non-Target) that are flashed pseudo-randomly to elicit the P300 response. EEG data were recorded using 32 active wet electrodes with three conditions: flash duration 50ms, 80ms or 110ms.

  *32 channels: FP1, FP2, AFz, F7, F3, F4, F8, FC5, FC1, FC2, FC6, T7, C3, Cz, C4, T8, CP5, CP1, CP2, CP6, P7, P3, Pz, P4, P8, PO7, O1, Oz, O2, PO8, PO9, PO10.*

- **Individual Imagery**: This dataset investigates the impact of mental tasks on the binary classification performance of BCI users with central nervous system (CNS) impairments, such as individuals with stroke or spinal cord injury (SCI). It provides EEG recordings from nine disabled participants (with either spinal cord injury or stroke), collected across two separate experimental sessions on different days. The participants performed five distinct mental tasks (MT) following a cue-guided experimental paradigm.

  *30 channels: AFz, F7, F3, Fz, F4, F8, FC3, FCz, FC4, T3, C3, Cz, C4, T4, CP3, CPz, CP4, P7, P5, P3, P1, Pz, P2, P4, P6, P8, PO3, PO4, O1, O2.*

- **Music BCI**: This dataset is used to investigate the applicability of musical stimuli within the P300 paradigm. Eleven participants listened to polyphonic music segments composed

of three instruments playing simultaneously. The researchers designed a multi-stream odd-ball paradigm, in which each instrument played a repetitive standard musical pattern inter-spersed with occasionally occurring deviant patterns. Participants were instructed to attend to one specific instrument while ignoring the other two.

*63 channels: Fp1, Fp2, AF7, AF3, AF4, AF8, F9, F7, F5, F3, F1, Fz, F2, F4, F6, F8, F10, FT7, FC5, FC3, FC1, FCz, FC2, FC4, FC6, FT8, T7, C5, C3, C1, Cz, C2, C4, C6, T8, TP7, CP5, CP3, CP1, CPz, CP2, CP4, CP6, TP8, P9, P7, P5, P3, P1, Pz, P2, P4, P6, P8, P10, PO7, PO3, POz, PO4, PO8, O1, Oz, O2.*

- **RSVP Speller**: This dataset is intended for the study of brain-computer interface (BCI) spellers based on Rapid Serial Visual Presentation (RSVP). Researchers investigated the RSVP speller under three experimental conditions, focusing on the effects of stimulus onset asynchrony (SOA) and the use of color features. A vocabulary consisting of 30 symbols was presented sequentially in a pseudo-random order at the same spatial location. Twelve participants took part in the experiment.

  *63 channels: Fp1, Fp2, AF3, AF4, F9, F7, F5, F3, F1, Fz, F2, F4, F6, F8, F10, FT7, FC5, FC3, FC1, FCz, FC2, FC4, FC6, FT8, T7, C5, C3, C1, Cz, C2, C4, C6, T8, TP7, CP5, CP3, CP1, CPz, CP2, CP4, CP6, TP8, P9, P7, P5, P3, P1, Pz, P2, P4, P6, P8, P10, PO9, PO7, PO3, POz, PO4, PO8, PO10, O1, Oz, O2.*

- **SEED/SEED-IV/SEED-VII**: This dataset series is designed for research on emotion recognition tasks and includes EEG signals corresponding to three, four, or seven cate-gories of emotions. Participants were exposed to emotionally charged film clips in a con-trolled laboratory environment to elicit specific emotional states. Each participant took part in three or four experimental sessions.

  *62 channels: Fp1, Fpz, Fp2, AF3, AF4, F7, F5, F3, F1, Fz, F2, F4, F6, F8, FT7, FC5, FC3, FC1, FCz, FC2, FC4, FC6, FT8, T7, C5, C3, C1, Cz, C2, C4, C6, T8, TP7, CP5, CP3, CP1, CPz, CP2, CP4, CP6, TP8, P7, P5, P3, P1, Pz, P2, P4, P6, P8, PO7, PO5, PO3, POz, PO4, PO6, PO8, CB1, O1, Oz, O2, CB2.*

- **SHU**: This dataset is intended for research on motor imagery tasks. Participants were in-structed to perform corresponding hand movement imagery based on directional cues dis-played on a screen, involving a classification task between left-hand and right-hand motor imagery.

  *32 channels: FP1, FP2, FZ, F3, F4, F7, F8, FC1, FC2, FC5, FC6, CZ, C3, C4, T7, T8, A1, A2, CP1, CP2, CP5, CP6, PZ, P3, P4, P7, P8, PO3, PO4, OZ, O1, O2.*

- **Monitoring ERP**: This dataset corresponds to an EEG experiment designed to elicit error-related potentials (ErrPs) under conditions where users monitor the behavior of an external system over which they have no control. During the experiment, participants sat in front of a computer screen displaying a moving cursor (represented by a green square) and a designated target location. The cursor's movement was autonomously controlled, and par-ticipants were instructed to observe the system's performance while being aware that the objective was for the cursor to reach the target.

  *64 channels: Fp1, AF7, AF3, F1, F3, F5, F7, FT7, FC5, FC3, FC1, C1, C3, C5, T7, TP7, CP5, CP3, CP1, P1, P3, P5, P7, P9, PO7, PO3, O1, Iz, Oz, POz, Pz, CPz, Fpz, Fp2, AF8, AF4, AFz, Fz, F2, F4, F6, F8, FT8, FC6, FC4, FC2, FCz, Cz, C2, C4, C6, T8, TP8, CP6, CP4, CP2, P2, P4, P6, P8, P10, PO8, PO4, O2.*

- **Motor imagery**: This data set consists of EEG data from 9 subjects. The cue-based BCI paradigm consisted of four different motor imagery tasks, namely the imagination of move-ment of the left hand (class 1), right hand (class 2), both feet (class 3), and tongue (class 4). Two sessions on different days were recorded for each subject. Each session is comprised of 6 runs separated by short breaks. One run consists of 48 trials (12 for each of the four possible classes), yielding a total of 288 trials per session.

  *22 channels: Fz, FC3, FC1, FCz, FC2, FC4, C5, C3, C1, Cz, C2, C4, C6, CP3, CP1, CPz, CP2, CP4, P1, Pz, P2, POz.*

We calculate the JS-similarity between the spatial features $U$ derived from different datasets used in the pre-training stage, which is defined as:

$$JS - similarity = \frac{1}{1 + JS}, \qquad JS = JS(P||Q) = \frac{1}{2}KL(P||M) + \frac{1}{2}KL(Q||M), \quad (11)$$

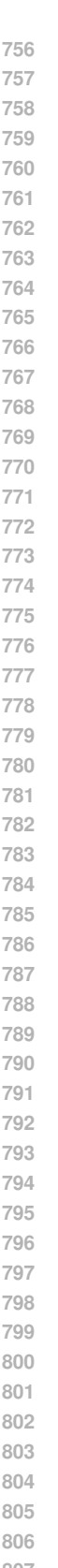
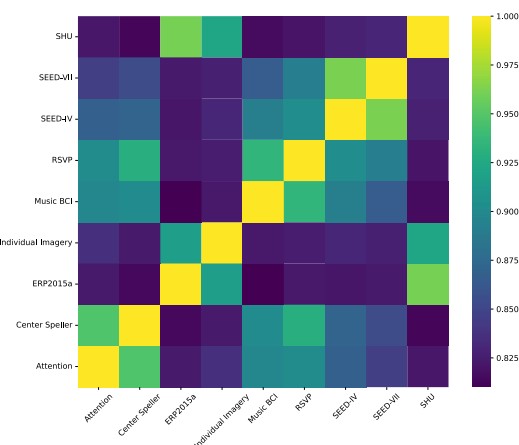

Figure 4: Distribution of data in the dataset employed for model pre-training.

where $M = \frac{1}{2}(P + Q)$ and $JS$ denotes the Jensen-Shannon Divergence (Lin, 1991) between distributions $P$ and $Q$, which is a symmetric and smoothed variant of the KL divergence. A higher JS-similarity indicates that the two data distributions are more similar, which is beneficial for training the diffusion model. As shown in Figure 4, the differences in data distribution primarily stem from the number of channels in the original data, rather than factors such as task type. Nevertheless, even in cases of the greatest distributional divergence, a similarity score exceeding 0.8 can still be achieved, indicating that such variation is unlikely to pose significant challenges for training diffusion models.

## A.3 BASELINES

- **Spherical Spline Interpolation (SSI)**: SSI is an interpolation method commonly used in EEG signal processing to reconstruct missing or corrupted electrode channels. It maps electrode positions onto the scalp surface and employs spherical spline functions to model the spatial relationships among electrodes, enabling smooth reconstruction of absent channels based on surrounding signals. Compared with simple linear or distance-weighted interpolation, SSI better preserves the spatial continuity and topographical characteristics of scalp potentials.

- **RePaint**: The RePaint model was originally proposed for image inpainting and also employs a diffusion-based framework. During the denoising stage, it replaces the noise in the visible regions with the noise added during the forward diffusion process, and utilizes a resampling technique to make the reconstructed image more similar to the ground truth. In our experiments, to adapt the model for EEG signals, we replaced the original 2D convolutional networks with 1D convolutional networks and, during sampling, substituted the noise in the visible channels with the corresponding noise from the forward diffusion process.

- **BioDiffusion**: BioDiffusion is a diffusion model specifically designed for biomedical signals and supports a range of tasks, including unconditional, label-conditional, and signal-conditional generation. In the signal-conditional generation, it enables the reconstruction of missing EEG segments using the visible portions of the signal. The forward diffusion process in BioDiffusion follows the standard procedure of typical diffusion models. During the backward diffusion process, the model takes as input a combination of the partially missing original signal and noise, which jointly guide the sampling process.

- **Gram**: Gram is a pre-trained general-purpose EEG model designed for both classification and reconstruction tasks. Inspired by Large Language Models (LLMs), it employs a two-stage framework to learn generalized representations from EEG signals while enabling efficient data repair. In the first stage, a temporal tokenizer encoder discretizes EEG signals into base classes, capturing essential temporal patterns. In the second stage, Gram leverages masked modeling to predict the base classes of corrupted EEG data using the retained data, followed by reconstructing the corrupted signals via a tokenizer decoder.

Table 4: Parameters and their values used in the model.

| (Hyper)parameter | Value(s) |
|---|---|
| Unet init dimension | 128 |
| Unet layers | 3 |
| Unet layer dimensions | [1, 2, 3] |
| Convolutional kernel size | 3 |
| Diffusion steps | 1000 |
| Pre-train epochs | 40 |
| Fine-tune epochs | 50 |
| Learning rate for pre-train/fine-tune | 5e-4 |
| Weight decay for pre-train/fine-tune | 1e-2 |
| Sampling steps | 100 |
| Batch size | 5096 |
| Optimizer for pre-train/fine-tune | AdamW |
| Scheduler for pre-train/fine-tune | CosineAnnealingLR |

## A.4 EVALUATION METRICS

- **Pearson Correlation Coefficient (PCC):** PCC is a statistical measure that evaluates the linear relationship between two variables. It ranges from -1 to 1, where 1 indicates a perfect positive correlation, -1 indicates a perfect negative correlation, and 0 means no linear correlation. PCC is widely used in data analysis to understand how strongly two variables are related.

- **Relative Root Mean Square Error (RRMSE):** RRMSE is a normalized version of the Root Mean Square Error (RMSE), used to measure the accuracy of a model's predictions. It is calculated by dividing the RMSE by the mean of the observed data, making it easier to compare errors across different datasets or scales. A lower RRMSE value indicates better predictive performance.

- **Peak Signal-to-Noise Ratio (PSNR):** PSNR is a widely used metric for measuring the quality of reconstructed or compressed images and videos. It compares the maximum signal value (e.g. 255) to the noise introduced by compression or distortion. A higher PSNR value generally indicates better image quality and less distortion. However, since EEG signals do not have a fixed maximum value, we use the original signal values as a substitute for the maximum.

- **Structural Similarity Index Measure (SSIM):** SSIM is a perceptual metric used to evaluate the similarity between two images. It considers changes in structural information, luminance, and contrast, making it more aligned with human visual perception. SSIM values range from -1 to 1, where 1 indicates perfect similarity. It is commonly used to assess image quality in compression, restoration, and enhancement tasks. When applying it to EEG signals, we manually compute the mean and variance of both the original and reconstructed data according to the formula definition, and subsequently calculate the SSIM value.

## A.5 MORE MODEL & EXPERIMENT DETAILS

In the experiments, we set the predefined maximum number of EEG channels $C_{max}$ to 95. The specific channel names are as follows:

*Fp1, Fpz, Fp2, Fp9, Fp10, Nz, AF1, AF2, AFz, AF3, AF4, AF5, AF6, AF7, AF8, AF9, AF10, F1, F2, Fz, F3, F4, F5, F6, F7, F8, F9, F10, FC1, FC2, FCz, FC3, FC4, FC5, FC6, FT7, FT8, FT9, FT10, C1, C2, Cz, C3, C4, C5, C6, T7, T8, T9, T10, I1, I2, CP1, CP2, CPz, CP3, CP4, CP5, CP6, TP7, TP8, TP9, TP10, P1, P2, Pz, P3, P4, P5, P6, P7, P8, P9, P10, PO1, PO2, POz, PO3, PO4, PO5, PO6, PO7, PO8, PO9, PO10, O1, O2, Oz, O9, O10, Iz, CB1, CB2, A1, A2.*

The electrode names and definitions are primarily based on the guidelines provided by the International Federation of Clinical Neurophysiology (IFCN) for the standard placement of scalp EEG electrodes (2017). In addition, the set includes several commonly used electrodes that are specific to certain public datasets (e.g., CB1, CB2). Table 4 presents the model architecture and the parameters required for training. All training and testing procedures are conducted using four NVIDIA L40 GPUs with Python 3.8.10 and PyTorch 2.3.1 + CUDA 12.2.

As noted in Section 3.4.2, it is essential to ensure a consistent channel order across all datasets. The reason is as follows: As illustrated in Figure 5, when two datasets share the same channel names but differ in channel order, the spatial features $U_1$ and $U_2$ obtained through direct forward mapping exhibit structurally identical formats. However, as shown in Equations 12 and 13, the specific feature elements at corresponding positions differ. This discrepancy has a significant impact on SVD, since by its nature, the leftmost column vectors in matrix $U$ correspond to the largest singular values and thus contain more 'information'. When channel orders are inconsistent, the information distribution in $U$ becomes misaligned, which may hinder effective model training.

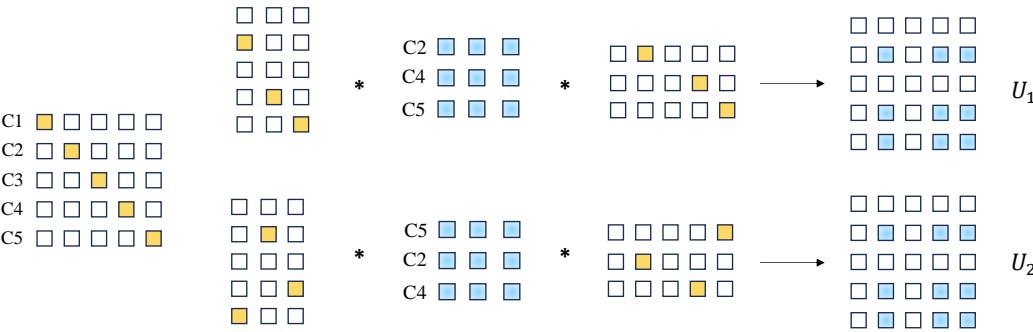

Figure 5: When different datasets share identical channel names but in varying orders, the resulting $U$ matrices appear structurally consistent; however, their underlying distributions differ.

$$U_1 = \begin{bmatrix} 0 & 0 & 0 \\ 1 & 0 & 0 \\ 0 & 0 & 0 \\ 0 & 1 & 0 \\ 0 & 0 & 1 \end{bmatrix} * \begin{bmatrix} c_{21} & c_{22} & c_{23} \\ c_{41} & c_{42} & c_{43} \\ c_{51} & c_{52} & c_{53} \end{bmatrix} * \begin{bmatrix} 0 & 1 & 0 & 0 & 0 \\ 0 & 0 & 0 & 1 & 0 \\ 0 & 0 & 0 & 0 & 1 \end{bmatrix} = \begin{bmatrix} 0 & 0 & 0 & 0 & 0 \\ 0 & c_{21} & 0 & c_{22} & c_{23} \\ 0 & 0 & 0 & 0 & 0 \\ 0 & c_{41} & 0 & c_{42} & c_{43} \\ 0 & c_{51} & 0 & c_{51} & c_{53} \end{bmatrix}, \quad (12)$$

$$U_2 = \begin{bmatrix} 0 & 0 & 0 \\ 0 & 1 & 0 \\ 0 & 0 & 0 \\ 0 & 0 & 1 \\ 1 & 0 & 0 \end{bmatrix} * \begin{bmatrix} c_{51} & c_{52} & c_{53} \\ c_{21} & c_{22} & c_{23} \\ c_{41} & c_{42} & c_{43} \end{bmatrix} * \begin{bmatrix} 0 & 0 & 0 & 0 & 1 \\ 0 & 1 & 0 & 0 & 0 \\ 0 & 0 & 0 & 1 & 0 \end{bmatrix} = \begin{bmatrix} 0 & 0 & 0 & 0 & 0 \\ 0 & c_{22} & 0 & c_{23} & c_{21} \\ 0 & 0 & 0 & 0 & 0 \\ 0 & c_{42} & 0 & c_{43} & c_{41} \\ 0 & c_{52} & 0 & c_{53} & c_{51} \end{bmatrix}. \quad (13)$$

## A.6 MORE EXPERIMENTAL RESULTS

This section presents the complete experimental results that could not be included in the main text of the paper.

### A.6.1 THE COMPLETE ABLATION RESULTS OF THE FORWARD & BACKWARD MAPPING ANALYSIS

Figure 6 presents the complete analysis results of forward & backward mappings across the three datasets. The trends observed in the PSNR and SSIM metrics are consistent with those of the PCC.

### A.6.2 THE COMPLETE ABLATION RESULTS OF LOSS FUNCTION

Table 5 illustrates the differences in reconstructed data of the diffusion-based baselines when trained with L1 and L2 loss functions. As indicated in the tables, the use of the L1 loss generally outperforms the L2 loss in most cases, with only a few exceptions.

### A.6.3 THE RESULTS OF REGION-SPECIFIC CHANNEL MISSING

We test the channel reconstruction quality when missing channels are concentrated in a specific brain region. The two datasets with a larger number of channels, SEED and Monitor ERP, are approximately divided into four brain regions: frontal, central, posterior, and lateral, with each region

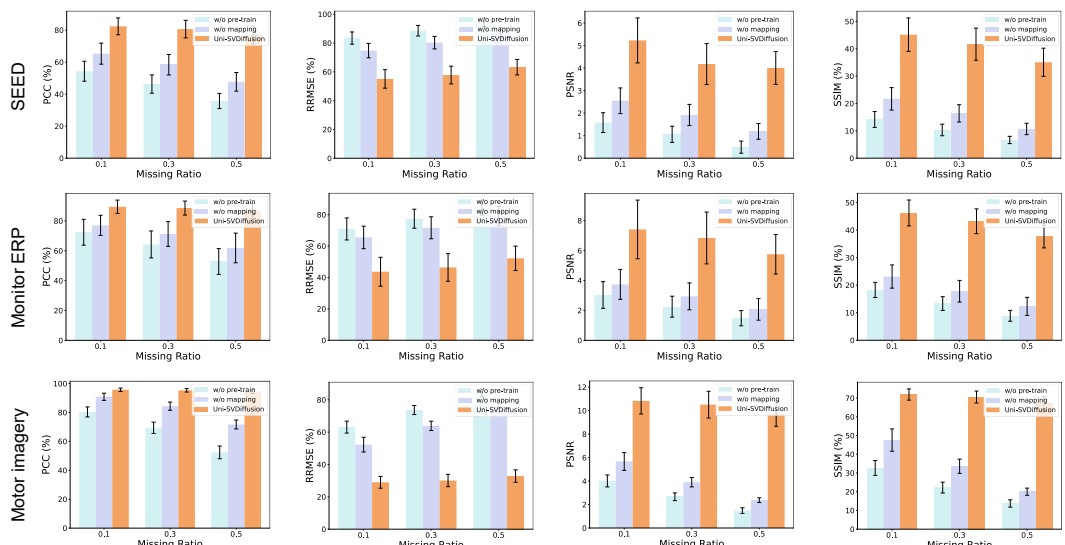

Figure 6: The complete results (mean/standard deviation) of the forward & backward mapping analysis on the three datasets.

Table 5: The loss function ablation results (mean/standard deviation) of the diffusion-based baselines on three datasets, with the best results highlighted in bold.

| Dataset | Method | Loss | 0.1 | | | | 0.3 | | | | 0.5 | | | |
|---|---|---|---|---|---|---|---|---|---|---|---|---|---|---|
| | | | PCC [%](↑) | RRMSE [%](↓) | PSNR (↑) | SSIM [%](↑) | PCC [%](↑) | RRMSE [%](↓) | PSNR (↑) | SSIM [%](↑) | PCC [%](↑) | RRMSE [%](↓) | PSNR (↑) | SSIM [%](↑) |
| SEED | RePaint | L1 | 57.65/6.93 | 84.25/02.90 | 1.49/0.30 | 11.61/2.82 | 49.35/5.92 | 87.23/2.50 | 1.19/0.25 | 9.03/2.07 | 40.39/5.04 | 90.97/2.16 | 0.82/0.21 | 6.51/1.55 |
| | | L2 | 55.02/07.67 | 86.16/03.02 | 1.30/0.31 | 9.50/2.71 | 44.15/06.24 | 89.82/02.51 | 0.94/0.24 | 6.85/1.78 | 34.34/05.18 | 93.89/02.21 | 0.55/0.21 | 4.76/1.29 |
| | BioDiffusion | L1 | -3.31/1.44 | 109.35/00.83 | -0.78/0.07 | 0.25/0.03 | -2.98/1.26 | 109.23/0.76 | -0.77/0.06 | 0.25/0.03 | -2.60/1.10 | 109.15/0.70 | -0.76/0.06 | 0.25/0.02 |
| | | L2 | -3.55/1.36 | 109.69/00.82 | -0.80/0.06 | 0.21/0.04 | -3.11/1.22 | 109.52/00.78 | -0.79/0.06 | 0.21/0.03 | -2.54/0.98 | 109.35/0.70 | -0.78/0.06 | 0.22/0.03 |
| Monitor ERP | RePaint | L1 | 78.48/5.37 | 66.24/5.40 | 3.61/0.72 | 22.80/6.01 | 77.09/4.68 | 67.69/4.62 | 3.41/0.61 | 22.30/5.03 | 72.48/4.72 | 72.40/4.62 | 2.82/0.57 | 18.46/5.09 |
| | | L2 | 66.50/06.11 | 83.26/02.98 | 1.60/0.31 | 8.35/1.08 | 53.09/04.43 | 87.85/01.82 | 1.13/0.18 | 5.53/0.93 | 39.92/03.49 | 91.67/01.45 | 0.76/0.14 | 3.93/0.64 |
| | BioDiffusion | L1 | 24.54/13.21 | 101.51/7.09 | -0.11/0.61 | 3.77/2.63 | 25.17/13.09 | 101.18/7.09 | -0.08/0.61 | 3.89/2.63 | 27.14/13.32 | 100.12/7.35 | 0.01/0.64 | 4.14/2.75 |
| | | L2 | 3.47/1.00 | 105.72/0.29 | -0.48/0.02 | 0.12/0.04 | 3.35/1.01 | 105.76/0.28 | -0.49/0.02 | 0.11/0.04 | 3.19/1.01 | 105.82/0.24 | -0.49/0.02 | 0.11/0.03 |
| Motor imagery | RePaint | L1 | 91.08/01.59 | 64.35/02.22 | 3.83/0.30 | 36.80/3.22 | 83.14/1.70 | 72.54/1.37 | 2.79/0.16 | 24.57/1.83 | 67.42/1.89 | 80.64/0.98 | 1.87/0.11 | 14.75/1.09 |
| | | L2 | 91.15/01.41 | 64.84/02.15 | 3.77/0.29 | 35.94/3.11 | 82.99/01.66 | 73.23/01.42 | 2.71/0.17 | 23.58/1.85 | 66.49/01.97 | 81.38/01.03 | 1.79/0.11 | 13.96/1.06 |
| | BioDiffusion | L1 | 6.84/2.49 | 103.19/0.90 | -0.27/0.08 | 0.25/0.05 | 6.78/2.61 | 103.20/0.93 | -0.27/0.08 | 0.26/0.06 | 6.36/2.59 | 103.36/0.95 | -0.29/0.08 | 0.26/0.06 |
| | | L2 | 8.11/2.46 | 102.60/0.82 | -0.22/0.07 | 0.31/0.07 | 8.01/2.49 | 102.63/0.81 | -0.23/0.07 | 0.30/0.07 | 7.65/2.47 | 102.76/0.81 | -0.24/0.07 | 0.29/0.06 |

Table 6: Reconstruction performance of SEED and Monitor ERP datasets under region-specific channel missing.

| Region | SEED | | | | Monitor ERP | | | |
|---|---|---|---|---|---|---|---|---|
| | PCC | RRMSE | PSNR | SSIM | PCC | RRMSE | PSNR | SSIM |
| Frontal | 67.10/8.48 | 72.93/6.28 | 3.42/0.73 | 28.89/4.92 | 85.73/3.03 | 49.34/4.46 | 6.17/0.79 | 43.89/5.30 |
| Central | 77.23/6.05 | 62.38/6.43 | 4.15/0.89 | 35.58/6.26 | 90.73/5.21 | 40.87/11.67 | 8.21/2.92 | 46.71/4.52 |
| Posterial | 75.27/6.73 | 64.57/5.91 | 3.83/0.76 | 33.91/5.68 | 87.03/6.55 | 48.03/10.22 | 6.57/1.86 | 41.25/5.75 |
| Lateral | 78.92/5.81 | 59.80/6.34 | 4.51/0.91 | 40.04/6.00 | 89.21/5.74 | 44.07/12.32 | 7.53/2.84 | 44.54/5.34 |

containing 12 channels. As shown in Table 6, in most cases, when the number of missing channels is the same, the model's reconstruction performance remains largely consistent and is comparable to that of random channel missing.

### A.6.4 THE COMPLETE ABLATION RESULTS OF THE SAMPLING STEPS

In this section, we investigate the impact of the number of sampling steps on the quality of the generated data. As illustrated in Figure 7(A), the PCC metric exhibits a steady increase followed by a plateau as the number of sampling steps increases. In contrast, the RRMSE metric reaches its optimal value within 100 to 150 sampling steps; further increasing the number of steps leads to a noticeable degradation in reconstruction performance. This indicates that as the number of sampling steps increases, the overall trend of the reconstructed signal can still closely follow that of the ground truth. However, distortions may occur in certain fine details (Figure 7(B)). The trends observed in PSNR and SSIM are generally consistent with the aforementioned results, as PSNR is closely related to the reconstruction error, while SSIM primarily reflects the overall structural similarity.

An examination of the RRMSE trends on the SEED and Monitor ERP datasets reveals that when the number of missing channels is relatively small, fewer sampling steps are required to achieve

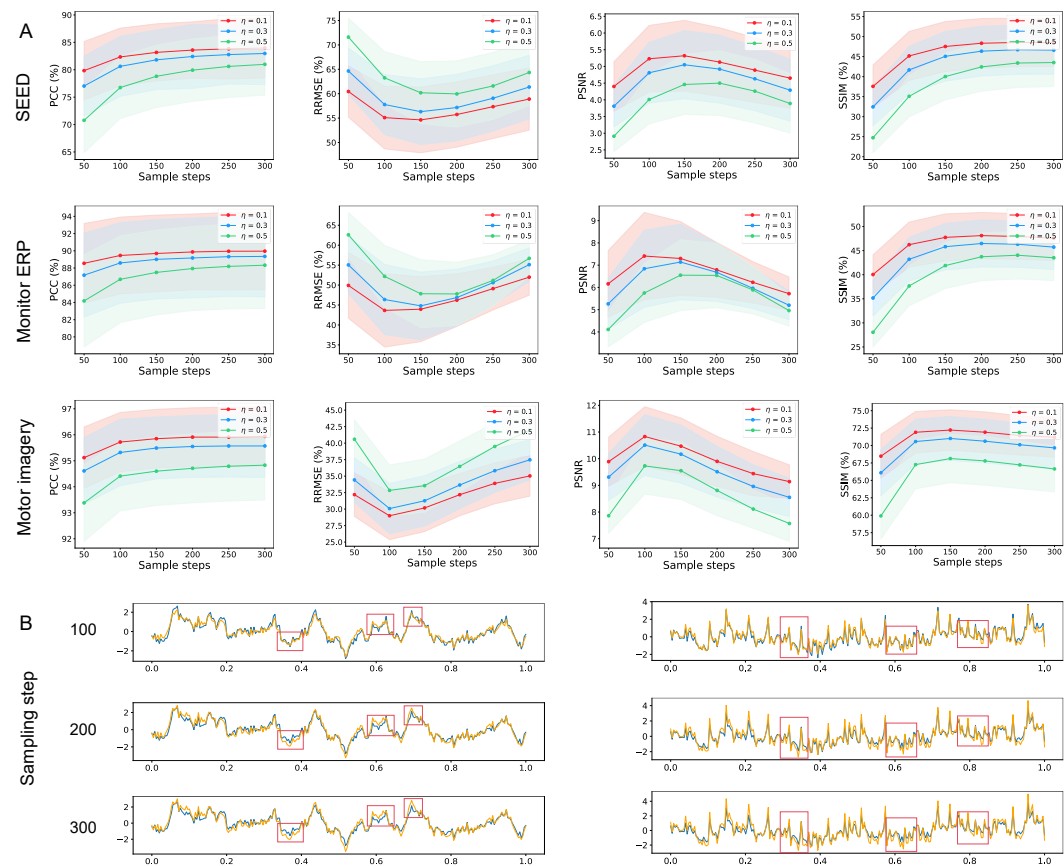

Figure 7: (A) Ablation results on the number of sampling steps across three datasets. (B) Using the Motor imagery dataset as an example, the reconstructed details (red frames) of the same channel vary with different sampling steps ($\eta = 0.3$).

optimal performance. Conversely, a greater number of missing channels necessitates more sampling steps. Based on this observation, we recommend dynamically adjusting the number of sampling steps according to the number of channels to be generated. For consistency and fair comparison across experiments presented in the main text, we uniformly adopt 100 sampling steps.

### A.6.5 MORE VISUALIZATION

Figures 8 through 10 present additional visualization results on the three datasets, with a channel missing ratio of 0.3.

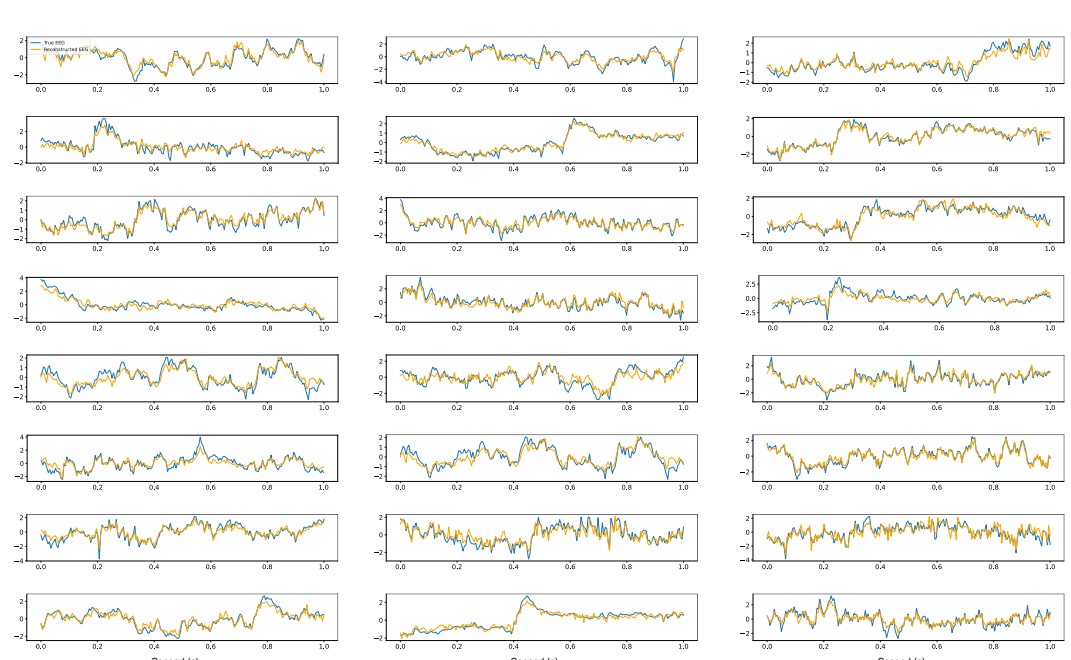

Figure 8: More visualization on the SEED dataset.

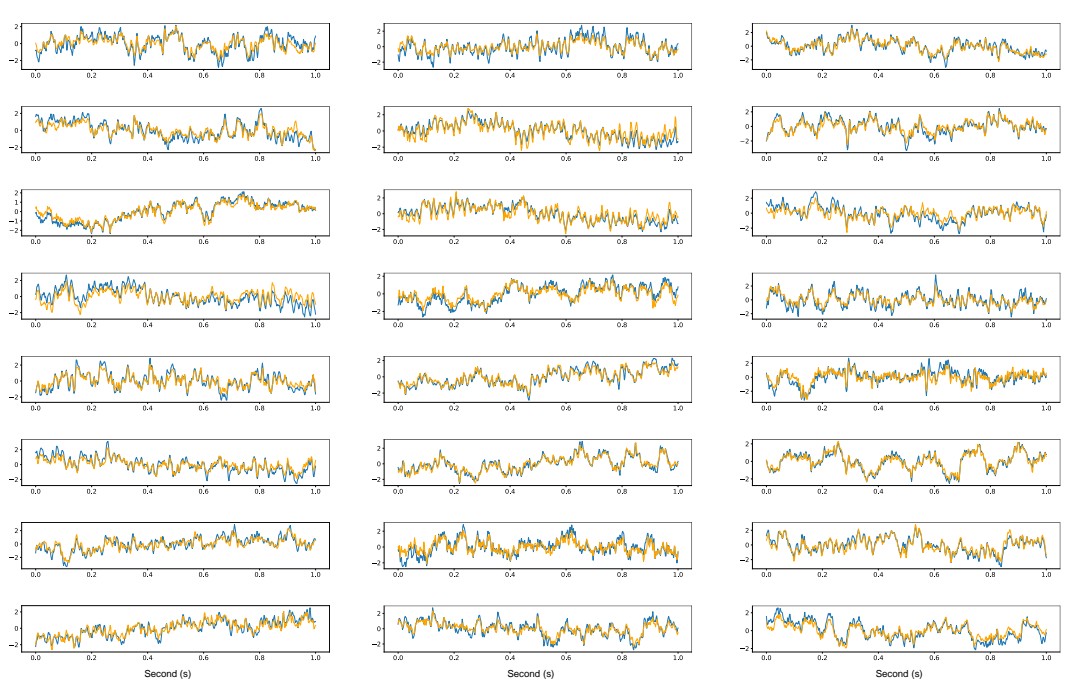

Figure 9: More visualization on the Monitor ERP dataset.

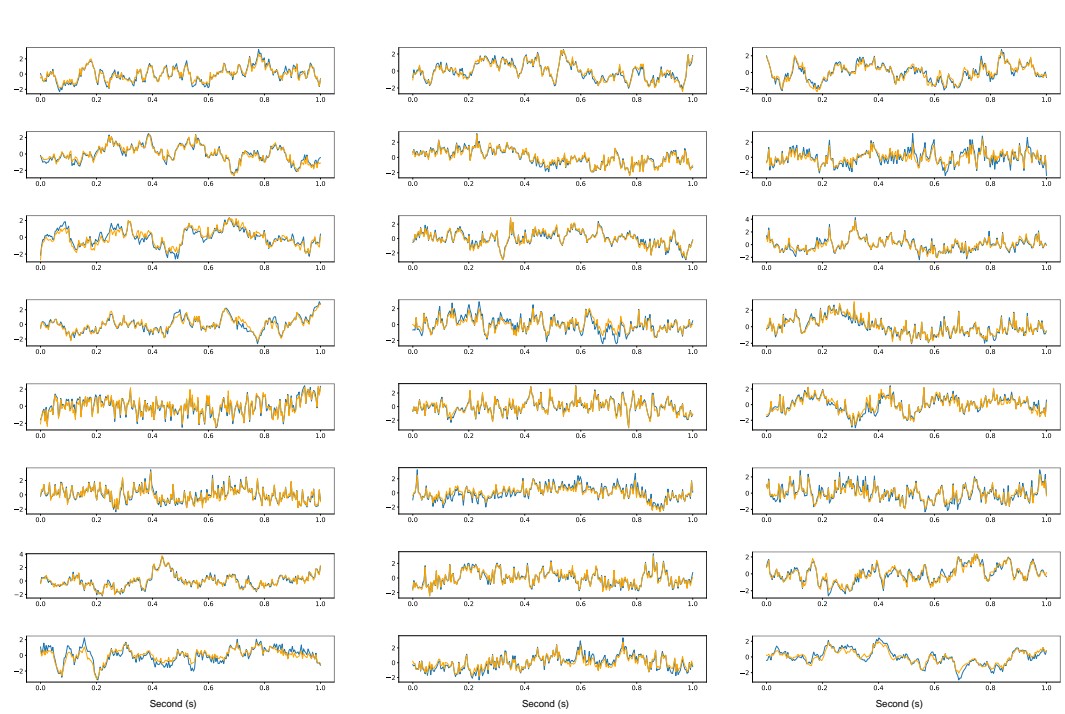

Figure 10: More visualization on the Motor imagery dataset.