# OpenReview forum: "Missing but Not Lost: Reconstructing Arbitrarily Missing EEG Channels with Pre-trained Diffusion"
_ICLR.cc/2026/Conference — Submitted to ICLR 2026_

### Official Review · Reviewer_AGAu · 2025-10-20

**Soundness:** 2
**Presentation:** 2
**Contribution:** 2
**Rating:** 2
**Confidence:** 4

**Summary:**

This paper addresses the issue of noisy EEG channels by using a diffusion model on the spatial SVD space of the corrupted data. Moreover, the diffusion happens in an expanded SVD space which enables training on datasets with different channel counts. The method is then evaluated on different datasets and assessed with reconstruction metrics.

**Strengths:**

- The ablation between L1 and L2 losses is very valuable. As a nit, I would say that it might be worth putting it in the appendix and mentioning it to justify the choice.
- The experimental setup with the different datasets is compelling.
- The idea of using the SVD space to do diffusion to abstract away the channel dimension is new and very interesting.
- Trying to adapt the diffusion training to datasets with different channel counts is a super interesting idea especially since the datasets have such low subject counts.

**Weaknesses:**

- **Code**: the readme in the code doesn’t provide instructions to obtain the figures (or clear python package requirements).
- **Figures**: the font size in the figures is very small, this makes them hard to read. If the figures were done with matplotlib a simple trick is to reduce the figsize -> this will make the font appear bigger.
 - **Downstream performance assessment**: the 2 baseline methods tailored to biosignals BioDiffusion [1] and Gram [2] both report downstream performance as the key measure of whether their reconstruction method works. Restoration/reconstruction metrics are seen as only safeguards/sanity checks to ensure that the method is working as intended. Indeed the “real-world applications” of EEG signals usually involve some sort of classification or regression. I think measuring performance of classifiers on the signals reconstructed with the different methods is ultimately the sole judge of whether a method is valuable. For this reason, the numbers presented here are without equivalent in the literature and it’s difficult to judge what their significance is. I think in order to completely ignore downstream performance assessment it would be useful to first make it clear that there is a strong correlation between reconstruction metrics and downstream performance in a first work.
- **Baseline**: The Gram paper [2] uses spherical splines as a baseline in Fig. 3. I think it makes sense to use this very naive baseline as it is I think the default method in applications.
- **Forward-backward mapping**: as highlighted above, I think being able to train on datasets with different channel counts is great. However, the explanation of how this happens in section 3.4 is in my opinion not clear. I think it deserves more rigor in defining $P$, denoting the dimensions and not going straight to an example or a figure.


[1] Li, X., Sakevych, M., Atkinson, G., and Metsis, V. Biodiffusion: A versatile diffusion model for biomedical signal synthesis. Bioengineering, 11(4):299, 2024.

[2] Li, Z., Zheng, W.-L., and Lu, B.-L. Gram: A large-scale general eeg model for raw data classification and restoration tasks. In ICASSP 2025-2025 IEEE International Conference on Acoustics, Speech and Signal Processing (ICASSP), pp. 1–5. IEEE, 2025.

**Questions:**

- Where does the Uni come from in Uni-SVDiffusion?
- “Compared to directly reconstructing the complete EEG signal, the spatial features have more stable format” -> what does stable mean here?
- In my understanding, the diffusion process here is essentially a function which goes from an incomplete $U$ to a completed $U$. Would it be possible to rather directly learn in a supervised learning manner this function and use this as a baseline? There are potentially really nice architectures that could be adapted for these matrix to matrix regression problems.

**Details Of Ethics Concerns:**

I reviewed the same exact paper for NeurIPS 2025 with small adjustments probably from the LLM use mentioned at the end of the paper.

---

### Official Review · Reviewer_PBz6 · 2025-11-01

**Soundness:** 2
**Presentation:** 2
**Contribution:** 3
**Rating:** 4
**Confidence:** 4

**Summary:**

## Summary

This paper proposes Uni-SVDiffusion for EEG channel reconstruction using SVD-based spatio-temporal disentanglement with a guided diffusion model. After synthesizing an expert human review (Danny) and a calibrated AI multi-reviewer panel assessment, the consensus is Borderline Reject (5.5/10).

---

## Strengths

1. **Clarity/Presentation Issues:** Both reviews independently identify severe accessibility problems
2. **Problem Significance:** The problem is genuine and important
3. **Conceptual Merit:** The core idea has merit and represents a meaningful contribution
4. **Methodological Soundness (with caveats):** The methodology has a solid foundation, but with critical gaps

---

## Weaknesses

1. **Unvalidated Core Assumption:** SVD's temporal/spatial attribution is argued from linear algebra, not proved physiologically No empirical validation that V captures temporal structure or U captures spatial topographies *Impact:* If the fundamental assumption is wrong, the entire approach collapses
2. **Fragile Theoretical Foundation:** Full-row rank assumption relies on "no direct electrode connectivity" and "quasi-random temporal variation" Volume conduction in EEG induces strong linear mixing, potentially violating assumptions *Impact:* Method may fail in practice if assumptions don't hold
3. **Unfair Baseline Comparisons:** RePaint and Bio-Diffusion lack architecture for multiple channel configurations Comparison confounds pretraining (forward/backward mapping) with architectural benefits (SVD decomposition) *Impact:* Cannot determine if improvements come from architecture or pretraining advantage
4. **Severe Clarity Issues:** Dense Section 3.4, complex Figure 1 not accessible Expert reviewer disagreement (5.5-9.5) diagnostic of real barriers *Impact:* Reproducibility compromised, limits adoption
5. **Incomplete Ablations:** Missing no-V-guidance / partial-guidance study Cannot isolate contribution of sharing V *Impact:* Cannot validate core design choice
6. **Ethics Omission:** No discussion of privacy, dual-use concerns for brain data reconstruction *Impact:* Insufficient broader impact consideration

**Strengths:**

1. **Clarity/Presentation Issues:** Both reviews independently identify severe accessibility problems
2. **Problem Significance:** The problem is genuine and important
3. **Conceptual Merit:** The core idea has merit and represents a meaningful contribution
4. **Methodological Soundness (with caveats):** The methodology has a solid foundation, but with critical gaps

**Weaknesses:**

* **Unvalidated Core Assumption:** SVD's temporal/spatial attribution is argued from linear algebra, not proved physiologically No empirical validation that V captures temporal structure or U captures spatial topographies *Impact:* If the fundamental assumption is wrong, the entire approach collapses
* **Fragile Theoretical Foundation:** Full-row rank assumption relies on "no direct electrode connectivity" and "quasi-random temporal variation" Volume conduction in EEG induces strong linear mixing, potentially violating assumptions *Impact:* Method may fail in practice if assumptions don't hold
* **Unfair Baseline Comparisons:** RePaint and Bio-Diffusion lack architecture for multiple channel configurations Comparison confounds pretraining (forward/backward mapping) with architectural benefits (SVD decomposition) *Impact:* Cannot determine if improvements come from architecture or pretraining advantage
* **Severe Clarity Issues:** Dense Section 3.4, complex Figure 1 not accessible Expert reviewer disagreement (5.5-9.5) diagnostic of real barriers *Impact:* Reproducibility compromised, limits adoption
* **Incomplete Ablations:** Missing no-V-guidance / partial-guidance study Cannot isolate contribution of sharing V *Impact:* Cannot validate core design choice
* **Ethics Omission:** No discussion of privacy, dual-use concerns for brain data reconstruction *Impact:* Insufficient broader impact consideration

**Questions:**

* **Validate SVD Temporal/Spatial Separation:** Provide empirical analysis showing V captures temporal structure (spectral decomposition, phase-amplitude) Demonstrate U corresponds to meaningful spatial topographies
* **Address Rank Assumption:** Report effective ranks / singular spectra across all datasets and window lengths Analyze condition numbers to assess numerical stability
* **Provide Fair Baseline Comparisons:** Option A: Compare non-pretrained Uni-SVD vs non-pretrained baselines (isolate architecture) Option B: Pretrain all models on same channel-size datasets (isolate pretraining benefit)
* **Add V-Guidance Ablation:** No-V-guidance / partial-guidance study to isolate contribution of sharing V Quantify how much of the performance gain comes from this design choice

---

### Official Review · Reviewer_ivaC · 2025-11-01

**Soundness:** 2
**Presentation:** 3
**Contribution:** 2
**Rating:** 2
**Confidence:** 4

**Summary:**

The authors present a diffusion framework for cross-channel EEG reconstruction that generates signals for missing channels. The approach combines SVD and a diffusion model.

**Strengths:**

-	The authors’ methodology is applicable for datasets of different sizes.
-	Several baselines tested.

**Weaknesses:**

-	I am skeptical of using interpolation or diffusion models to reconstruct missing EEG data. Since the goal of EEG is to analyze the neural activity of a specific subject—whether for diagnosing a disorder or for a BCI application—replacing actual measurements with generated data is methodologically unsound. The generated data, no matter how realistic, provides no new information about that individual.

   Fundamentally, this process is equivalent to adding sophisticated noise to the dataset. The model ensures this noise resembles a typical EEG signal, but it remains a fabrication that obscures the original, subject-specific information we seek to capture.


-	SVD is used for decades in EEG data processing. Too much space of the paper is paid to this fairly basic topic.
-	On the other hand, the actual diffusion model was not really presented and discussed.
-	Section 3.4.2 presents material not quite clear. It would be much better to the reader if you could you follow earlier introduced notations C_V, C_M, C_max. Also “P.S.” and “We recommend” does not look appropriate for an academic paper.

**Questions:**

See weaknesses

---

### Official Review · Reviewer_XJBk · 2025-11-01

**Soundness:** 3
**Presentation:** 3
**Contribution:** 3
**Rating:** 6
**Confidence:** 3

**Summary:**

The paper proposes Uni‑SVDiffusion, a pre‑trained diffusion framework to reconstruct arbitrarily missing EEG channels. The key idea is to factor an EEG segment with SVD into “spatial” and “temporal” parts, train a diffusion model on the spatial part, and at test time guide sampling with temporal features computed from the visible channels. A forward/backward channel‑mapping using a fixed identity basis allows joint pre‑training across datasets with different montages. Experiments pre‑train on nine datasets and evaluate on three downstream sets with up to 64 channels.

**Strengths:**

- SVD-based separation into spatial/temporal EEG features, enabling effective guided reconstruction.

- Identity-basis channel mapping enabling unified pre-training across datasets with varying montages.

- Empirical results demonstrating significant performance improvements over prominent baselines.

**Weaknesses:**

- Unproven Stability of V: The assumption that V from partial EEG is stable is unsubstantiated. A quantitative analysis comparing full vs partial V is necessary.
- Baseline Fairness: Poor baseline performance (e.g., BioDiffusion) suggests possible setup/configuration issues. Include stronger baselines (graph-aware methods, spatial regression).
- Ignoring Montage Geometry: The proposed mapping disregards electrode geometry; experiments comparing geometry-aware methods or shuffled channel orders are recommended.

**Questions:**

Did experiments include shuffling channel orders or geometry-aware models for comparison?

---

### Official Review · Reviewer_NdMW · 2025-11-02

**Soundness:** 3
**Presentation:** 3
**Contribution:** 3
**Rating:** 4
**Confidence:** 3

**Summary:**

Uni-SVDiffusion reconstructs missing EEG channels by running a diffusion model on SVD-derived spatial features while conditioning on temporal features computed from visible channels. A forward/backward identity-based mapping unifies heterogeneous montages for cross-dataset pre-training. Across multiple datasets and missing ratios, the method outperforms spline interpolation, two diffusion inpainting baselines, and a pretrained EEG model, with useful zero-shot behavior. Ablations favor L1 loss, show mapping requires pre-training, and suggest an optimal range of sampling steps. Arbitrary-length inference degrades smoothly.

**Strengths:**

Clear motivation for channel dropout; elegant temporal-spatial factorization; simple montage-unifying mapping; strong empirical results with zero-shot utility; reasonable ablations; downstream classification validation.

**Weaknesses:**

The forward/backward mapping assumes consistent channel naming and precise reordering; the consequences of mislabels or electrode shifts are only discussed in an appendix. Evaluation metrics are mostly image-domain (PSNR, SSIM) or linear similarity; spectral and ERP-centric measures are missing. Baselines exclude prior learning-based channel super-resolution methods designed specifically for virtual electrode recovery.

**Questions:**

How are subject splits enforced in pre-training and fine-tuning to avoid leakage, and is any dataset present in both stages for the same subjects?

---

### Meta-Review · Area_Chair_CzvM · 2026-01-01

**Summary:**

Reviewers acknowledged the importance of the problem and the novelty of using diffusion with SVD-based representations, but raised consistent concerns about uncertain core assumptions (validity of SVD spatial/temporal separation), fairness and completeness of baselines, insufficient ablations to isolate key design choices, and clarity/presentation issues that hinder reproducibility. Several reviewers also questioned the practical and scientific validity of reconstructing missing EEG channels and the lack of broader impact or ethics discussion. The authors did not participate in the rebuttal.

**Reviewer Concerns:**

The authors did not participate in the rebuttal.

**Reviewer Scores:**

The authors did not participate in the rebuttal.

---

### Decision · Program_Chairs · 2026-01-26

Reject